# Fiscal Policies on New Passenger Cars in Europe: Implications for the Competitiveness of Electric Cars

Romeo Danielis 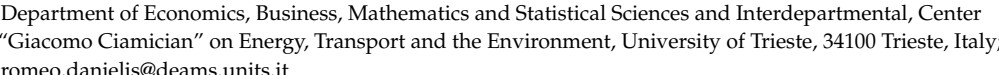

Department of Economics, Business, Mathematics and Statistical Sciences and Interdepartmental, Center "Giacomo Ciamician" on Energy, Transport and the Environment, University of Trieste, 34100 Trieste, Italy; romeo.danielis@deams.units.it

**Abstract:** The objective of this article is to review the fiscal policies applied to new passenger cars in 30 European countries. The fiscal policies considered include the value added tax, the vehicle registration tax, the purchase subsidy, the ownership tax, and the tax on fuels/electricity. The article illustrates their properties and their implementation in each country. In order to appreciate how the different national approaches translate into financial incentives/disincentives in relation to electric cars, each country's fiscal policies were applied to the Tesla Model 3 and the Toyota Corolla. The resulting acquisition costs and fiscal burden were then calculated and compared with reference to the year 2023. The results indicate that in some countries the Tesla Model 3 is cost competitive already in the acquisition phase (up to EUR 8524 cheaper), while in others is much more expensive (up to EUR 6590). The difference in the fiscal burden between the two car models ranges from EUR 448 to EUR 16,022, depending on the country. These findings have strong implications for social welfare, state budgets sustainability, and the need for car fiscal policy adjustments in the European countries.

**Keywords:** car fiscal policies; registration tax; ownership tax; subsidies; electric vehicles

## 1. Introduction

The reduction of local (PM, $NO_x$, $O_3$) and global ($CO_2$ eq.) pollutants emissions is a widely shared objective. Electric vehicles can significantly contribute to achieve such an objective. Technological advances in batteries and electric motors, investments by car manufacturers and, last but not least, the policies adopted at European level (directives on emission standards, on non-zero emission vehicles, etc.) are fostering the diffusion of electric vehicles. However, the uptake of electric vehicles occurs at very different paces, with Northern European countries leading the way and Southern and Eastern European ones lagging behind (Figure 1).

Several factors might explain such differences. They include the income level of the population, given that electric vehicles are even more expensive than their traditional counterparts. The current characteristics of the electric vehicles available on the market usually belong to higher car segments (luxury cars, SUV), while in some countries (e.g., in Italy see [1]) the most purchased vehicles are small–medium car segments. There is also an insufficient development of the charging infrastructure network. Furthermore, cultural factors might play a role, for example, the attachment to thermal vehicles, the mistrust regarding the life and safety of batteries, the belief that electric vehicles are not superior to thermal vehicles from an environmental point of view, or the fear that electric cars would lead to economic dependency (e.g., from China), and so on [2]. Finally, yet importantly, the slow electric car uptake could depend on the fiscal and regulatory policies implemented at national level.

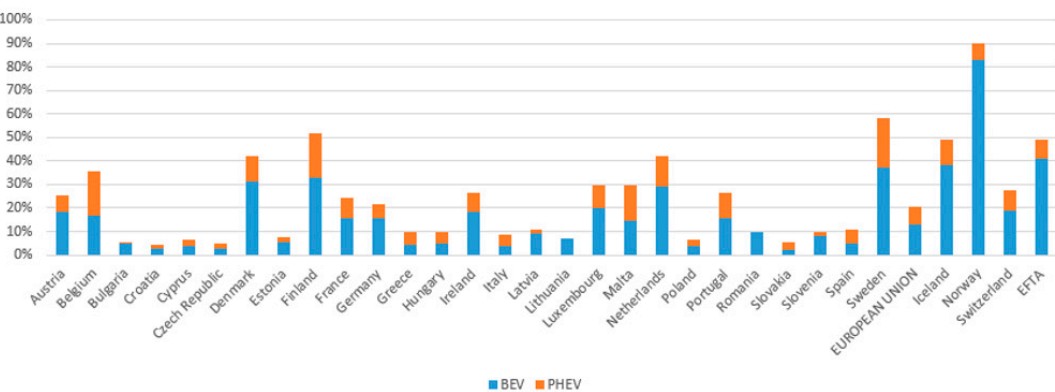

**Figure 1.** Market shares of BEVs and PHEVs in the first half of 2023. Source: [3,4].

This paper focuses on this last factor, with special attention on the fiscal policies on new passenger cars. The paper has two main objectives: (i) to illustrate and compare the fiscal policies applied in Europe at a national level (30 countries are considered); and (ii) to estimate their impact on the car acquisition costs and the fiscal burden of two successful cars models—the Tesla Model 3 and the Toyota Corolla—chosen as representatives of the electric (BEV) and petrol hybrid cars (HEV) since they are comparable in size and widely popular in their segment. In 2022 in Europe, 129,667 Toyota Corollas (19th ranking most sold car across all segments) and 121,610 Model 3s (second only to the Model Y among BEVs) were registered.

The fiscal policies considered were the value added tax (VAT) on car acquisition, the registration tax, the purchase subsidy, the ownership/circulation tax (In this paper, the term "ownership tax" is used in accordance with [3], although the term "circulation tax" is also quite common. Different countries might use different names: e.g., car tax, vehicle excise duty, etc.), and taxes on fuel/electricity consumption. Special attention is paid to the criteria used to calculate the registration and ownership tax given their impact on the car choice among different powertrains. After a brief introduction to the types and rationale of the vehicles' fiscal policies (Section 2), the paper illustrates the fiscal policies currently in force in 30 European countries (Section 3), taking advantage of two reports recently made available by the European Automobile Manufacturers' Association (ACEA) [3,4].

The application of the country-specific fiscal policies to the Tesla Model 3 and the Toyota Corolla (Section 4) allows readers to appreciate (i) the level of heterogeneity of the car fiscal policy strategies applied in each country; (ii) their impact on BEV cost competitiveness; (iii) how the fiscal tax burden is distributed between BEVs and internal combustion engine cars (ICEVs).

The state of vehicle fiscal policies is largely debated in the literature [5,6], with special reference to European countries; it is regularly illustrated, but not commented upon, by the ACEA reports [3,4]. On the contrary, a detailed discussion and analysis of the recent state of the vehicle fiscal policies is presented by [7]. Differently from [7], this paper includes the value added tax, an important component of the acquisition costs and of the overall tax burden.

## 2. Vehicle Fiscal Policies: A Premise

Vehicle fiscal policies are a complex topic for many reasons. First of all, it should be highlighted that there are many types of vehicles (i.e., bike, scooters, cars, vans, trucks; just considering road vehicles only), used to transport passengers, goods, or both, powered by different sources (petrol, diesel, methane, liquid propane gas, pure electric, hybrid with or without a plug). Depending on the type of vehicles, their use and power source, the fiscal policies might be quite different.

Vehicle fiscal policies can further differentiate among users: the major distinction is between private and business users. The latter use a vehicle within a production process and

are therefore subject to specific tax rules regarding the VAT regime, deductions, depreciation, and fringe benefits. Quite distinct business users are, for instance, car rental companies. Furthermore, there may be separate rules and taxes for some special users such as the disabled, large families, and those who carry out diplomatic services or public utility services (doctors, veterinarians, and paramedics). Finally, some policies—typically tax incentives or ownership taxes—may take into account the income bracket of the vehicle owner or differ according to the place of residence (provinces, regions, departments, etc.).

In this paper, the discussion is limited to cars bought and used by private individuals for private use, which represent a large share of the car market.

Fiscal measures can be applied in the different phases of a vehicle's life:

- At the time of purchase: via a value added tax, a vehicle registration tax, or a purchase subsidy;
- During the use phase: via a periodical ownership tax, a road distance-based tax (e.g., highway toll, truck toll), a congestion charge or a tax on vehicle fuels;
- When the vehicle is parked in specific areas;
- At the time of the resale and disposal of the vehicle: via specific fiscal measures.

There are various motivations for introducing fiscal policies, including:

- Funding transport infrastructure without using funds collected outside the transport sector;
- Promoting or deterring specific transport modes (public vs. private, rail vs. car, etc.);
- Managing traffic;
- Internalizing external costs associated with congestion, noise, or local and global pollutants.

There is much debate, both academically and on social media, about what the best vehicle fiscal policies are from the point of view of their effectiveness, efficiency, and fairness. There is strong empirical evidence that fiscal policies have a significant impact on the type of purchased car [8]. According to many studies [9,10], taxes paid at the time of acquisition are the most environmentally effective, but not necessarily the optimal ones [11]. In contrast, taxes that increase the expected costs over the entire life cycle of the car such as road taxes or fuel taxes are found to have less impact on the decision of which car to buy [12] because consumers tend to be "short-sighted" and calculate expected savings only in the first few years [13]. Another interesting result is that fiscal policies in the first decade of the 21st century mainly favored diesel cars at the expense of large petrol cars, due to their lower $CO_2$ emissions [12,13], despite the potential increase in NOx emissions [14,15].

Finding the right balance between policies is not an easy task, especially considering the fact that vehicle fiscal policies are the result of heated negotiations between interest groups and their political representatives.

## 3. Vehicle Tax Policies in Europe: An Overview

Table 1 summarizes the factors taken into consideration to calculate the vehicle registration tax and the ownership tax. The table has been prepared based on the recent ACEA reports [3,4] and national sites. Table 1 includes the 27 countries belonging to the European Union with the addition of Iceland, Norway, and the United Kingdom. Switzerland, which has different fiscal policies depending on the cantons, is not included.

It can be observed that VAT rates on vehicles are quite differentiated among countries. The lowest rate is 17% in Luxembourg, followed by Malta with 18%, and 19% in Cyprus, Germany, and Romania. The highest VAT rate is charged in Hungary (27%), followed by Denmark, Sweden, and Norway with 25%. However, one should not draw fast conclusions (for example, the higher the VAT, the greater the disincentive to purchasing vehicles), since each country then has complex rules of income tax reduction or deduction based on the VAT paid for vehicles for different categories of users, e.g., business users.

**Table 1.** The factors on which vehicle tax policies are based.

| Country | VAT | Registration Tax | Ownership Tax |
|---|---|---|---|
| Austria | 20% | Fuel consumption/$CO_2$ emissions (if >123 g/km in 2023) and purchase price + possible malus (if >175 g/km $CO_2$ in 2023). Limits are reduced year after year. BEVs exempt. | Engine power (kW) and $CO_2$ emissions. |
| Belgium | 21% | Displacement and age (Brussels-Capital); fuel, age, emission standard and $CO_2$ (Flanders); engine capacity, age, and $CO_2$-based bonus/malus system (Wallonia) | Displacement, $CO_2$ emissions, fuel type, and emission standards |
| Bulgaria | 20% | License plate cost (25 BGN) + fixed eco tax (160 BGN) | Engine power (kW), year of production, and emission standard |
| Croatia | 25% | Vehicle price, $CO_2$ emissions, and fuel type | Engine power (kW) and vehicle age |
| Cyprus | 19% | CO2 emissions (if >120 g/km) and displacement | $CO_2$ emissions |
| Czech Republic | 21% | Registration tax (max CZK 800) + eco tax based on emission standards (only if less than Euro3) | Private vehicles are not taxed |
| Denmark | 25% | Type of fuel, price of the car, and $CO_2$ emissions | $CO_2$ emissions |
| Estonia | 20% | Registration sheet (EUR 62) + registration card (EUR 130) fixed | Vehicles are not taxed |
| Finland | 24% | Vehicle price and $CO_2$ emissions | $CO_2$ emissions, vehicle weight, vehicle type |
| France | 20% | Varies by region + bonus (for EV and PHEV)/penalty based on $CO_2$ (if >125 g/km) and weight (BEVs exempt from weight penalty) | Fiscal power (CV), $CO_2$ emissions, weight |
| Germany | 19% | Fixed registration fee (EUR 26.30) | $CO_2$ emissions and displacement |
| Greece | 24% | Vehicle price and $CO_2$ emissions | Displacement or $CO_2$ emissions |
| Hungary | 27% | Displacement and emission standard (EURO) | Capacity and year of production |
| Ireland | 23% | Vehicle price, $NO_x$, and $CO_2$ emissions | Displacement or $CO_2$ emissions |
| Italy | 22% | Power + fixed fee (EUR 151 up to 53 kW) + power (if HP > 53 kW) | Engine power (kW), emission standard (EURO). BEV exempt for 5–10 years |
| Latvia | 21% | Registration costs (EUR 43.93) + national resources tax (EUR 55). BEV exempt | $CO_2$ emissions |
| Lithuania | 21% | Fixed registration fee (EUR 14.48). BEV exempt | Vehicles are not taxed |
| Luxembourg | 17% | Fixed registration fee (EUR 50), BEV (EUR 25) | $CO_2$ emissions |
| Malta | 18% | Vehicle price, emission standard (EURO), $CO_2$ emissions, and length | $CO_2$ emissions and age |
| Netherlands | 21% | $CO_2$ emissions | Weight, province, type of fuel, $CO_2$ emissions |
| Poland | 23% | Fixed tax (PLN 180). BEVs exempt | Vehicles are not taxed |
| Portugal | 23% | Displacement, $CO_2$ emissions + fixed registration tax (EUR 55) and license plate (EUR 45). BEV exempt | Displacement, type of fuel, $CO_2$ emissions. BEV exempt |
| Romania | 19% | Fixed registration fee (RON 40) | Engine capacity (cc) + road tax |
| Slovakia | 20% | Engine power and age + license plate cost (EUR 16.50) | Displacement. BEV exempt |
| Slovenia | 22% | Fuel type, $CO_2$ emissions, engine power, EURO emission standard. | Vignette on main roads (EUR 110 in 2023) |
| Spain | 21% | $CO_2$ emissions (if >120 g/km) | Engine power |

**Table 1.** *Cont.*

| Country | VAT | Registration Tax | Ownership Tax |
|---|---|---|---|
| Sweden | 25% | No registration fee on purchase | $CO_2$ emissions, vehicle weight, and fuel type |
| Iceland | 24% | 0% VAT on BEV | Weight and $CO_2$ emissions |
| Norway | 25% | 0% VAT on BEV. $CO_2$ emissions, $NO_x$ emissions, vehicle weight, air conditioning. Disposal tax | Replaced by insurance tax based on weight and fuel type |
| United Kingdom | 20% | Fixed amount | $CO_2$ emissions and fuel type |

It is quite common but not universal to calculate the registration and/or the ownership tax based on the $CO_2$ emissions during the use phase, clearly inspired by European decarbonization policies. A number of countries apply such a criterion for both taxes (Austria, Belgium, Cyprus, Denmark, Finland, France, Greece, Ireland, Malta, Netherlands, and Portugal). Some countries apply it only to the registration tax (Croatia, Slovenia, Spain, and Norway), others only to the ownership tax (Germany, Latvia, Luxembourg, Sweden, Iceland, and United Kingdom). The remaining group do not apply the $CO_2$ emissions criterion for either of the two taxes (Bulgaria, Czech Republic, Estonia, Hungary, Italy, Lithuania, Poland, Romania, and Slovakia), using instead the (local) air emission standards or the traditional engine displacement criterion. However, this does not mean that vehicles with zero $CO_2$ emissions are not favored. In several countries, they are exempt from one or both taxes. In Italy, for instance, BEVs are exempt from ownership taxes for at least 5 years (for 10 years in some regions) or subject to preferential taxation.

Regarding the registration tax, in several cases the acquisition cost of the vehicle net of the VAT is taken into account (Austria, Croatia, Denmark, Finland, Greece, Ireland, and Malta), either directly or as information to build a tax base to be subject to taxation. The intention is evidently to include elements of progressiveness in the registration tax, taking into account the fact that larger vehicles take up more space, are heavier, and less efficient.

In some (few) cases, however, the registration fee is fixed and relatively low. This is true in Estonia, Germany, Lithuania, Luxembourg, Poland, Romania, and Sweden. The presence of many Eastern European countries could suggest the need not to prevent access to private vehicles in countries coming from collective transport systems. Germany and Sweden, however, do not fit this consideration. In these cases, the interpretation could be traced back to the presence of a strong national automotive industry. However, this last observation must also be carefully evaluated as the registration tax is only one of the levers available to transport policy makers; subsidies and ownership taxes being the other instruments. Therefore, the evaluation must consider all available instruments and their actual use.

The cases of Iceland and Norway are striking since these two countries fully exempt BEVs from VAT, thus introducing a strong element of incentive. Furthermore, in the case of Norway, the registration tax takes into account $NO_x$ emissions (along with Ireland alone), in addition to the weight and the environmental tax linked to air conditioning.

Shifting the attention to ownership tax, it can be noted there is a frequent use of the engine power (measured in HP or $cm^3$) and the vehicle weight (in relation to the deterioration of the road surface) criterion, in addition to the $CO_2$ emissions or local air emissions standard known as EURO classification. In some cases, the tax takes the nature of a vignette (Slovenia and Romania), linked to the possibility of accessing main roads (motorways or expressways). In many countries, the application of the ownership tax is delegated to local authorities (departments, regions, provinces, etc.). In some countries (Estonia, Czech Republic, Lithuania, and Poland), ownership taxes have not yet been introduced.

A second ACEA publication [4] summarizes the incentives available to electric vehicles. In addition to the already examined registration or ownership taxes (and related exemptions), the document reports updated data on subsidies for the purchase of electric

vehicles and charging infrastructures (e.g., wall-box), both for private individuals and business users.

Based on these two documents and information deriving from national websites, Section 4 presents a comparison between the various components of the acquisition costs and the fiscal burden of two of the best-selling cars in Europe: a fully electric one and one equipped with a hybrid petrol power engine.

## 4. The Application of Fiscal Policies to Two Specific Car Models: Description of the Methodology and Results

To appreciate the impact of the fiscal policies enacted in the different countries, two of the most popular models were selected—the Tesla Model 3 (BEV) and the Toyota Corolla (HEV)—with similar but not equivalent characteristics, as it can be seen in Table 2.

**Table 2.** The characteristics of the Tesla Model 3 (BEV) and the Toyota Corolla (HEV).

| Car Model | Tesla Model 3 Standard Range Plus | Toyota Corolla 2.0 Hybrid Active |
| --- | --- | --- |
| Powertrain | BEV | HEV |
| MSRP (without VAT) | 34,818 | 29,344 |
| $CO_2$ (g/km) | 0 | 118 |
| HP (kW) | 239 | 135 |
| NOx (g/km) | 0 | 0.0051 |
| Weight | 1611 | 1340 |
| Engine displacement ($cm^3$) | | 1987 |
| Fuel efficiency | 14.7 kWh/100 km | 5.2 L/100 km |

The starting point of the comparison was the MSRP of the two car models. Since the purpose of this paper was to compare the fiscal policies enacted in the various countries, it was assumed that the MSRPs were the same across the countries and corresponded to the MSRPs accessed on 25 July 2023 on "https://alvolante.it/", an Italian website specializing in the Italian car market. Such an assumption is evidently a simplification because of the following:

- It does not consider that car manufacturers/car retailers might apply different price differentiation strategies across countries to take into account the difference in consumers preferences and purchasing power;
- It does not consider that car prices vary over time to respond to the competition or to adjust to cost changes;
- Importantly, the car manufacturers/car retailers might adjust their MSRPs to respond to fiscal policies. For instance, if a certain government grants a subsidy on BEVs, they might increase their MSRP to capture some of the fiscal stimulus. Alternatively, in some countries regulations are enacted that require car manufacturers/car retailers to supplement the government subsidy with car their own discounts.

Based on this paper's assumption, each country's value added tax rates were applied to the two car models. As noted above, Norway and Iceland, for the time being, waive the VAT to BEVs only.

### 4.1. Registration Tax

The registration tax is calculated assuming that the buyer is a private individual resident in the country.

As it can be seen from Figure 2, with few exceptions (e.g., Ireland, Norway, Italy), registration costs are waived for the Tesla Model 3. On the contrary, in most countries the Toyota Corolla does need to pay registration costs. The registration cost is particularly

high in Denmark (EUR 11,145), Norway (EUR 8111), Portugal (EUR 5863), and Greece (EUR 5416). These are the countries which have chosen (with the exception of Greece) to link the registration tax to the price of the vehicle and its environmental characteristics. In these countries, the simultaneous decision not to tax (or to tax to a lesser extent) the Tesla Model 3 means that the latter becomes very competitive. This is not the case in Italy. In fact, Italy is the only country in which the registration tax on the Tesla Model 3 is higher than that on the Toyota Corolla. The registration tax is calculated on the power resulting from the car's technical manual (In the case of Italy, for the Tesla Model 3 we assumed 53 kW of power and applied the lowest rate (3.51) with the addition of the fixed fee of EUR 15). For electric cars, the power considered is not the instantaneous power but the continuous power (i.e., during 30 min).

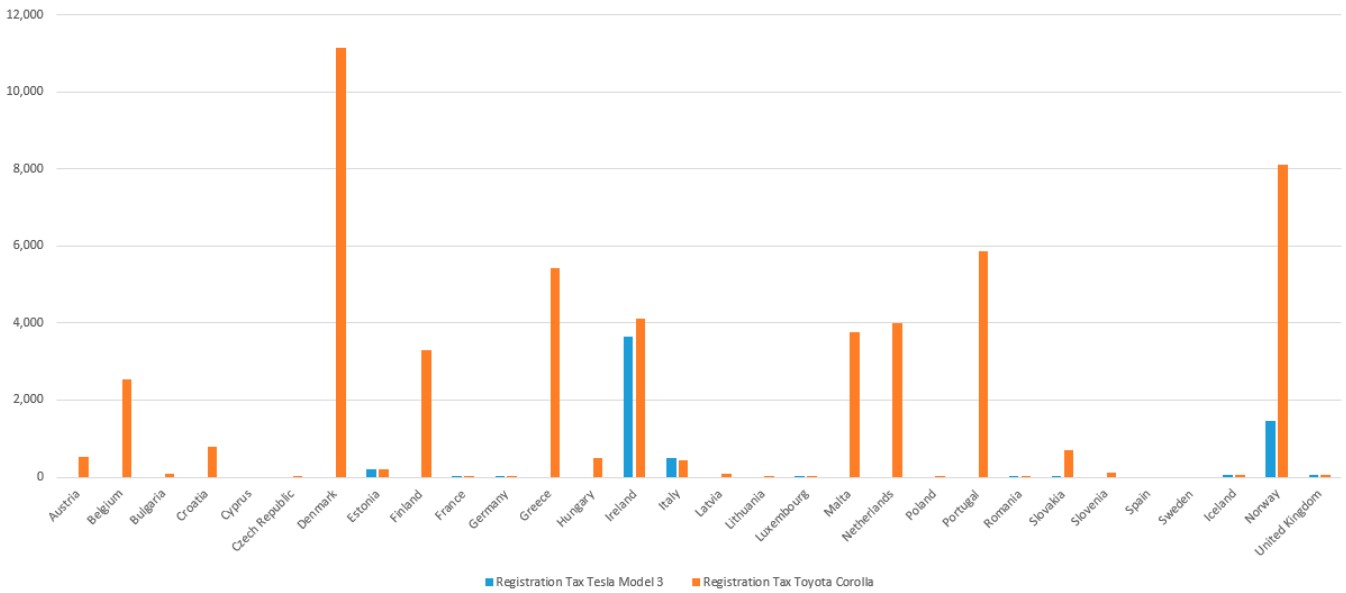

**Figure 2.** Registration tax applied to the two car models.

### 4.2. Subsidy

Almost all countries grant subsidies for purchasing of electric cars, with the exception of Bulgaria, the Czech Republic, Denmark, Latvia, the Netherlands, Slovakia, Iceland, and Norway (Figure 3). The presence of Denmark, Netherlands, Iceland, and Norway is interesting since, as discussed above, they are countries with a high penetration of electric cars. The most generous countries in granting incentives are Cyprus (EUR 12,000), Romania (EUR 11,500), Malta (EUR 11,000), Croatia (EUR 9291), Greece (EUR 8000), Germany (EUR 6750), and Sweden (EUR 6020). These values, however, should be taken with caution as these amounts could be available to a limited number of individuals [7]. In fact, to understand their real potential in promoting the spread of electric cars, not only their amount but the total funds available for each year should be considered.

### 4.3. Acquisition Costs

The acquisition costs were defined as the sum of the MSRP, the VAT, and the registration tax minus the subsidy for the two car models. The resulting amounts are illustrated in Figure 4.

It can be seen that, under the assumption of our model, in 10 countries the acquisition costs of the Tesla Model 3 are lower than those of the Toyota Corolla. Ordered according to the magnitude of the difference in the acquisition costs, these countries are as follows:, Norway, Malta, Greece, Cyprus, Romania, Denmark, Croatia, Portugal, Iceland and Germany.

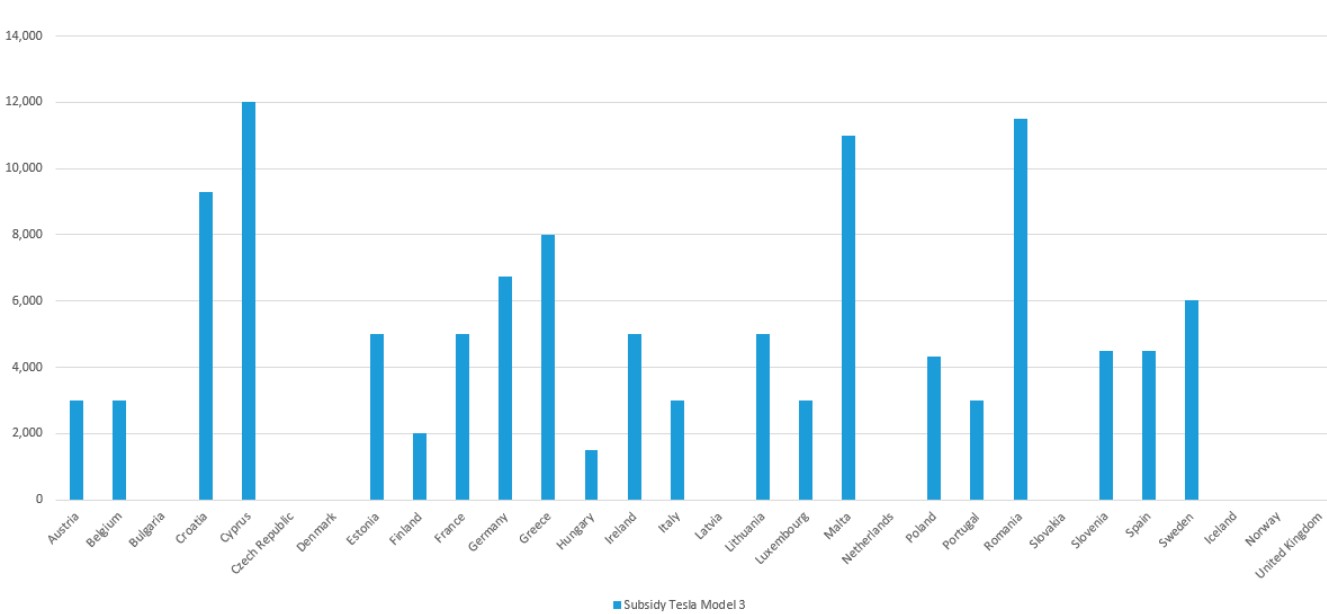

**Figure 3.** Subsidy granted to the Tesla Model 3.

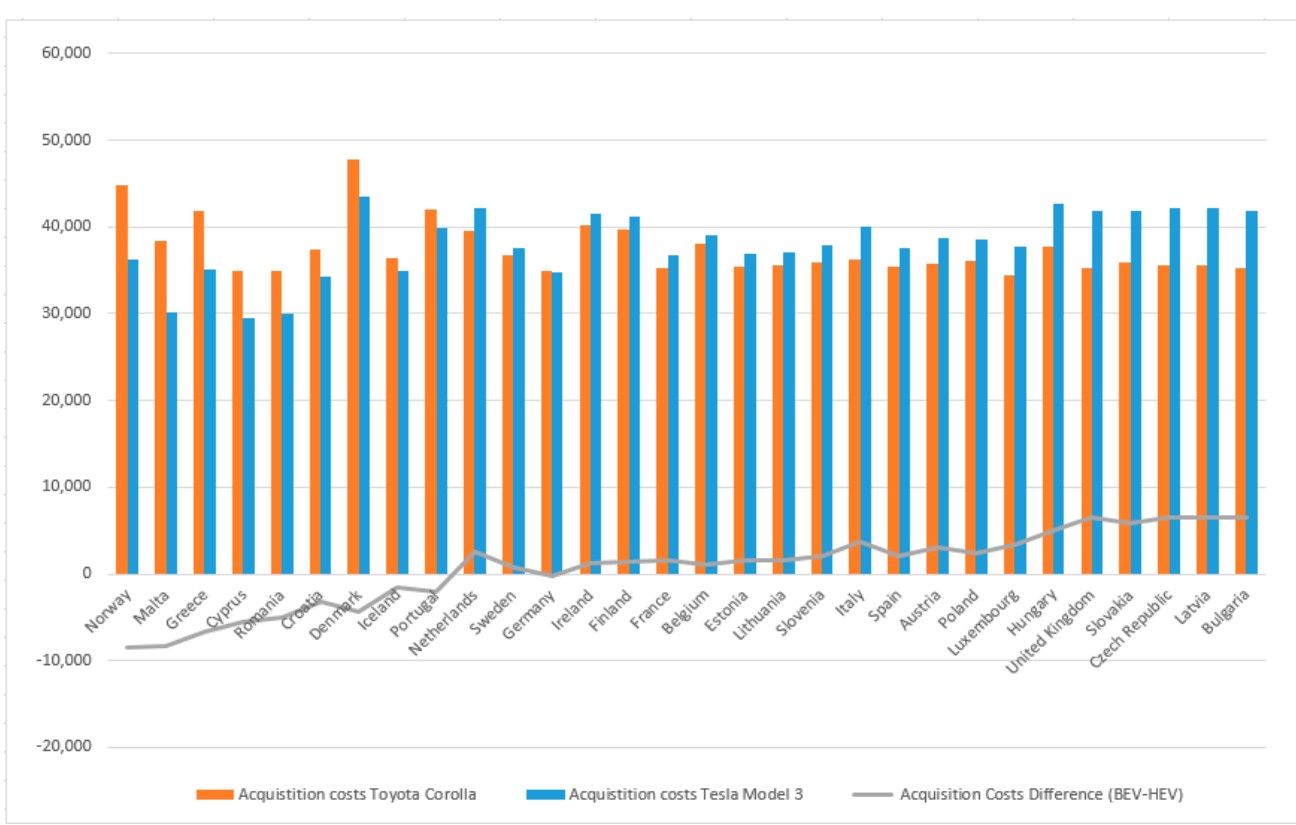

**Figure 4.** Acquisition costs of the two car models.

As discussed above, the main caveats in interpreting these results are the following: (i) for comparability purposes, the same MSRP across countries was assumed, and (ii) the countries granting very generous subsidies (Cyprus, Romania, Malta, Croatia, Greece, Germany, Sweden) have sufficient available public funds.

Notwithstanding these caveats, two observations can be made. First, the very different fiscal policies applied in these countries leads to very different BEV vs. HEV competitiveness conditions. The difference between the Tesla Model 3 and the Toyota Corolla varies

from EUR −8524 to 6590. Even without considering the savings on ownership tax, fuel costs, and maintenance costs over the lifetime of the car, the Tesla Model 3 is competitive in 10 countries. In other countries, the acquisition costs could be higher but compensated by operating cost gains. In the remaining countries, the Tesla Model 3 requires much higher upfront costs that could hardly be compensated by operating cost savings.

Second, in some countries—e.g., Denmark, Norway, Portugal, and Greece—the acquisition cost of the Toyota Corolla is very high, more than EUR 40,000. These are the countries with the highest registration taxes, based on $CO_2$ (and $NO_x$) emissions and the list price of the car. In Cyprus and Romania, instead, the acquisition cost of the Toyota Corolla is low and the cost competitiveness of the Tesla Model 3 is mainly due to the generous subsidy. In another group of countries (Denmark, Hungary, Netherlands, Latvia, Czech Republic, United Kingdom, Slovakia, Bulgaria, Ireland, Finland, and Italy), the acquisition cost of the Tesla Model 3 is above EUR 40,000. Consequently, in most of these countries, the Tesla Model 3 is not competitive, with the exception of Denmark where the acquisition cost of the Toyota Corolla is still higher.

*4.4. Ownership Tax*

In 22 out of the 30 countries, the Tesla Model 3 does not require payment of ownership tax (Figure 5). The tax is levied in Finland, Ireland, Spain, Slovenia, Malta, Sweden, Luxembourg, and Italy (In Italy, the exemption for electric cars varies from region to region, both in terms of duration (10 or 5 years) and amount (zero or 25%). The imputed value relates to 5 years of exemption and the remaining 3 years of tax reduction by a quarter.). On the contrary, the Toyota Corolla is charged an ownership tax. In an 8-year period, the ownership taxes due by an owner of the Toyota Corolla are highest in Austria (EUR 734), Netherlands (EUR 720), Italy (EUR 436), and Sweden (EUR 330).

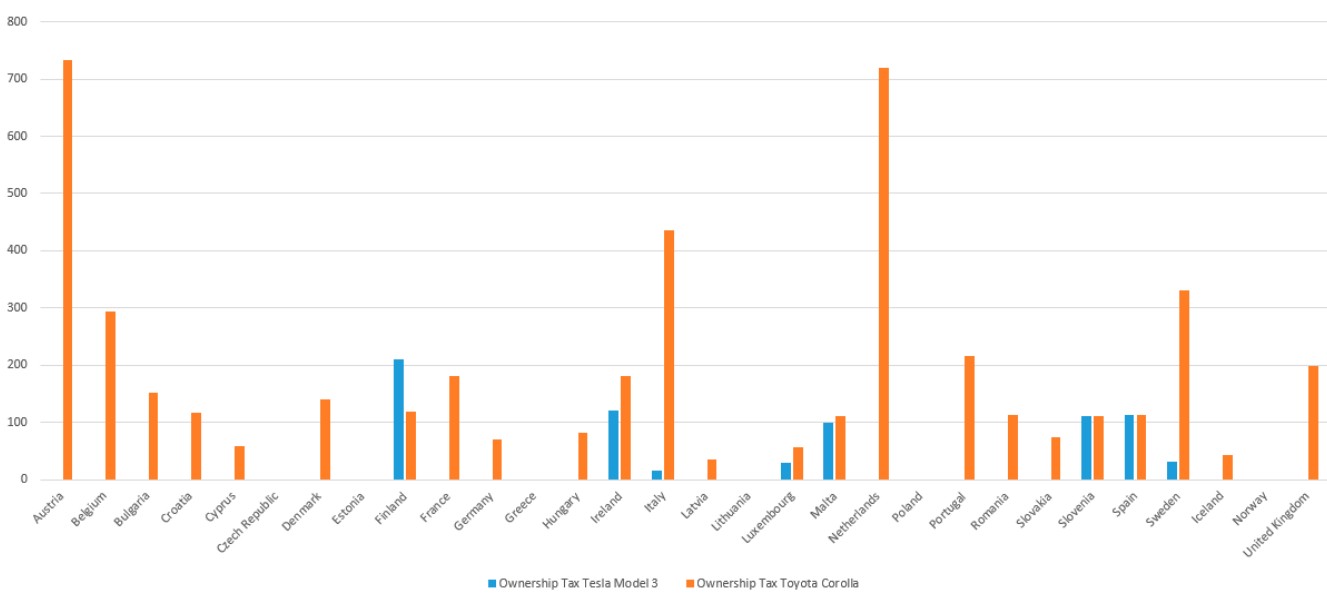

**Figure 5.** Ownership tax paid by the two car models during an 8-year period.

*4.5. Tax on Petrol/Electricity Consumption*

In order to estimate the tax on petrol/electricity consumption, the information on the excise duties on petrol published in [3] was used, and integrated with information found in national websites for the non-EU countries. Concerning the tax on electricity, the estimate was based on data from the Eurostat database, contained in the tables on "Electricity prices for household consumers—bi-annual data". Along the lines suggested by [7], the tax rate was calculated as the difference between the electricity prices "All taxes and levies included" and "Excluding taxes and levies", for the energy consumption bracket from 2500 kWh to

4999 kWh—band DC. The electricity prices did not include VAT. The prices referred to the first semester 2021 (before the price hike associated with the Ukrainian war), with the exception of the UK for which only the 2020, second semester, electricity price was available. It was assumed that a car travels 10,000 km per year in each country. For comparability purposes, the annual travel distance was kept constant across countries, although it might have been an overestimation for the smaller countries and an underestimation for the large ones. The fuel/electricity efficiency was the one communicated by the car manufacturer (Table 2). The results are illustrated in Figure 6.

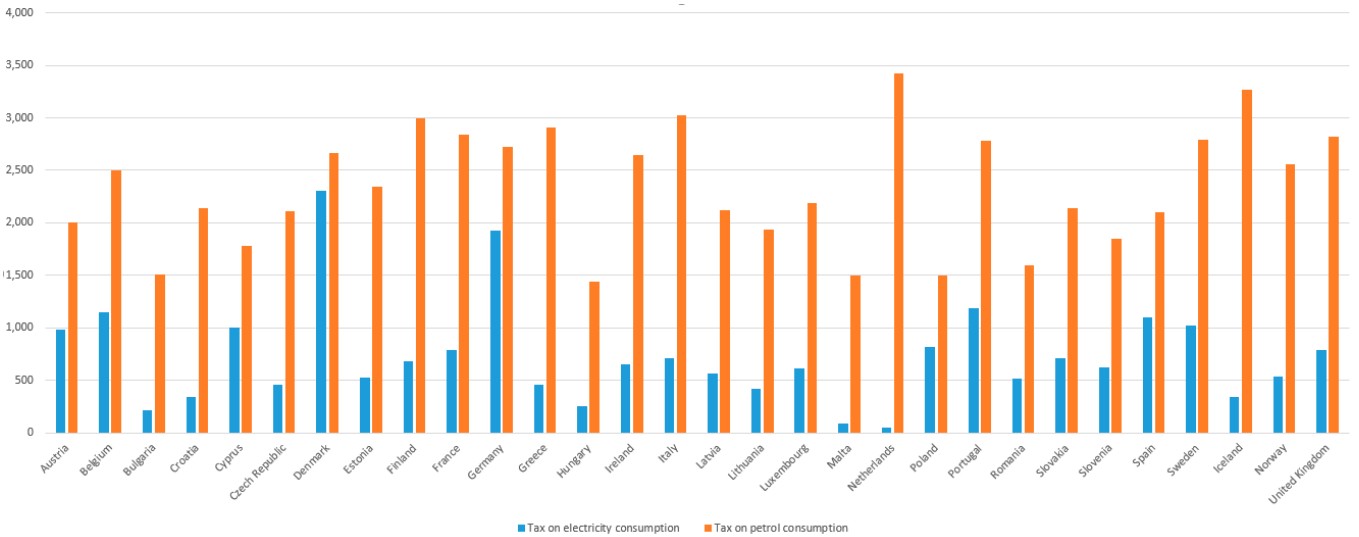

**Figure 6.** Taxes on petrol/electricity consumption paid by the two car models during an 8-year period.

It can be observed that in all countries, the taxes that a Toyota Corolla user would pay on petrol are higher than that a Tesla Model 3 user would pay on electricity. In some countries, such as Netherlands, Iceland, Greece, Italy, Finland, France, United Kingdom, and Norway, the fuel tax savings from driving a Toyota Corolla over 8 years would be EUR 2000 higher than driving a Tesla Model 3, up to 3121 euros in the Netherlands, which has a very low electricity tax.

### 4.6. Overall Fiscal Burden

Finally, the overall fiscal burden was calculated as the sum of the value added tax, the registration tax, the ownership tax, and the fuel/electricity tax over an 8-year period for the two car models (Figure 7).

Figure 7 offers a number of interesting insights:

- The current fiscal burden is higher on the Toyota Corolla than on the Tesla Model 3 in all the 30 countries considered.
- The difference between the fiscal burden on the Toyota Corolla and the Tesla Model 3 is higher than 10,000 euros in some countries: Norway (EUR 16,022), Malta (EUR 15,178), Greece (EUR 14,560), Cyprus (EUR 11,800), Romania (EUR 11,655), Croatia (EUR 10,639), Denmark (EUR 10,274), and Iceland (EUR 10,013).
- In Malta, Cyprus, Romania, and Croatia the fiscal burden on the Tesla Model 3 is negative because the subsidies are higher than the overall taxation (value added tax, registration tax, ownership tax, and tax on electricity). However, these results must be interpreted with care since such large subsidies might be granted on a low number of BEVs.
- The fiscal burden difference is very high in Norway because of the high registration taxes on non-BEVs (mainly $CO_2$ and $NO_x$ taxes) and the simultaneous VAT exemption; similar considerations apply to Iceland. Denmark is a special case because both the Toyota Corolla and the Tesla Model 3 are subject to a high fiscal burden, although

higher on the former due to the very high registration costs (mainly due to the $CO_2$ emissions tax).

- The fiscal burden difference is lower than 5000 euros in Slovenia (EUR 4636), Italy (EUR 4475), Spain (EUR 4344), Austria (EUR 4190), Poland (EUR 3775), Luxembourg (EUR 3688), Hungary (EUR 1785), United Kingdom (EUR 1136), Slovakia (EUR 1074), Czech Republic (EUR 540), Latvia (EUR 534), and Bulgaria (EUR 448).

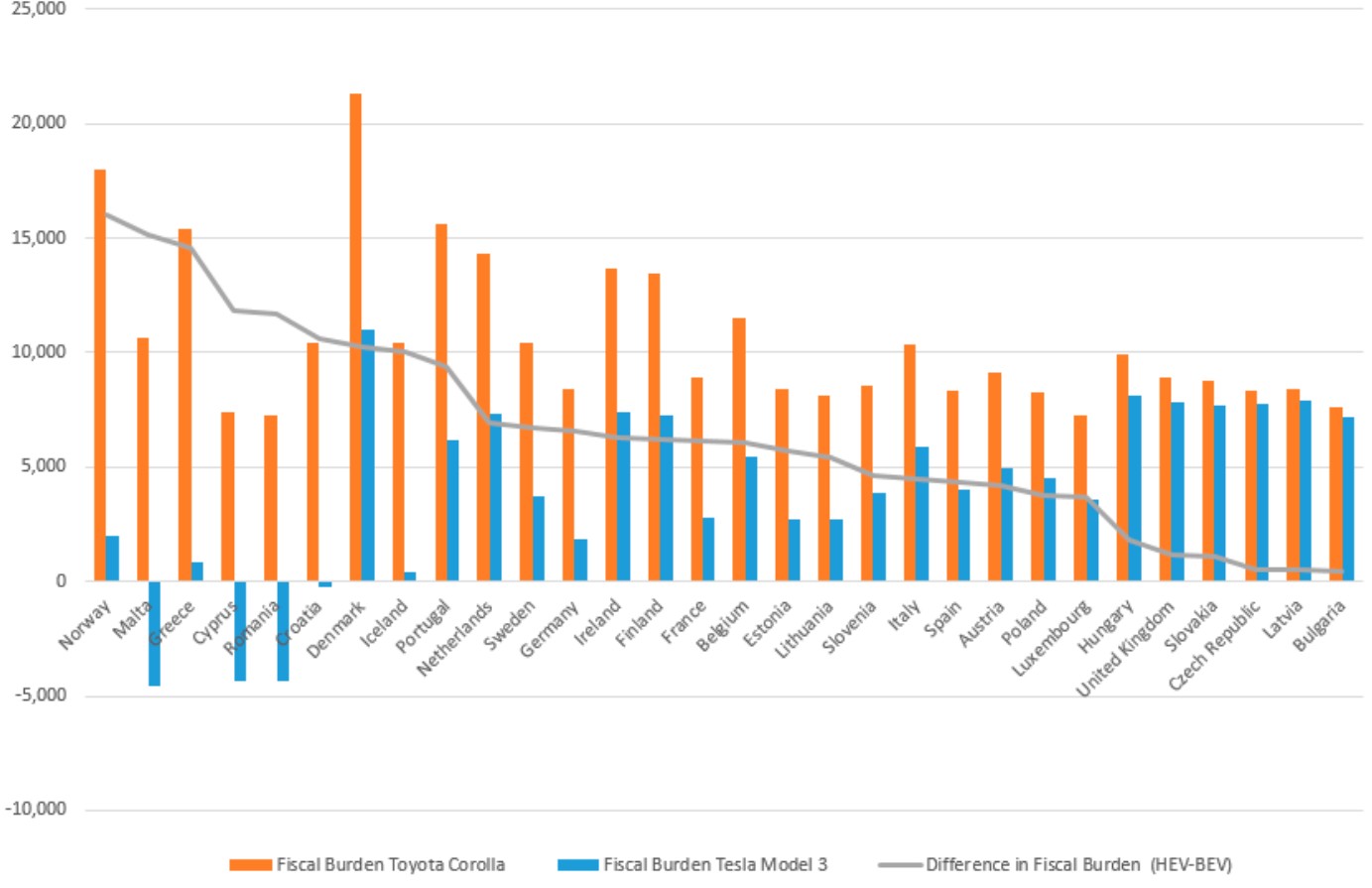

**Figure 7.** Fiscal burden on the two car models during and 8-year period.

Our results are in line with those found by [7]. They also underline that the net tax differential is achieved in some countries by subsidizing BEVs, while in others by imposing a large tax burden on polluting cars. The authors of [7] found that this holds true for both compact cars and SUVs.

## 5. Discussion, Limitations of the Study, and Implications for Electric Vehicle Incentive Policies

This paper reviewed the fiscal policies applied to new passenger cars in Europe. Having highlighted the complexity of the topic due to the many power sources used to power a car, the various type of users (private, business, renting companies, special users) and the utilization of income or place of residence criteria in determining the tax base, this paper focused on the main type of fiscal policies. It considered the value added tax, the vehicle registration tax, the purchase subsidy, the ownership tax, and the tax on fuels/electricity.

A caveat is that the paper did not take into account road distance-based tax, the congestion charge, or the parking fee or other fees associated with the resale and disposal of the car. Leaving out some of these taxes might be relevant in some cases. For instance, the congestion charge and the parking fee within large cities might further incentivize the

adoption of BEVs (for instance, the congestion charge in Milan and London, or the parking fee in Norway). Distance-based taxes, on the contrary, might be set irrespective of the type of powertrain (e.g., the highway toll in Italy).

The detailed analysis of the value added tax, the vehicle registration tax, the purchase subsidy, and the ownership tax in 30 European countries revealed very different approaches among countries. Concerning the VAT, the range varied from 17% to 27%, but with specific regards to BEVs, two countries, Norway and Iceland, decided to waive BEVs from such a charge, at least up to now (September 2023). A variety of criteria have been used for calculating the registration and ownership tax. A distinction has been made between those countries that differentiate such taxes based on CO2 emissions and those that do not. Concerning purchase subsidies, some countries grant very generous incentives, while others offer zero or very modest ones.

In order to appreciate how the different national approaches translate into financial incentives/disincentives, each country's fiscal policy was applied to two successful and representative car models, the Tesla Model 3 and the Toyota Corolla, and the acquisition costs and the fiscal burden were calculated. The results show that, under the assumption that the list price is the same across countries, the impact of the national fiscal policies is very strong, making the Tesla Model 3 cost competitive in some countries even in the acquisition phase (up to EUR 8524 cheaper), while in other countries much more expensive (up to EUR 6590). Next, the total fiscal burden was evaluated, adding the ownership tax and the tax on petrol/electricity consumption to the acquisition costs. The results indicate that in all countries, the fiscal burden on the Tesla Model 3 is smaller than that on the Toyota Corolla, but with huge variations among the countries. The fiscal burden difference between the two car models ranges from EUR 448 to 16,022. The addition of other fiscal policies left out of consideration—such as distance-based charges, congestion charges, or parking fees—is not likely to alter the conclusion that current fiscal policies, with different degrees, favors BEVs with respects to ICEVs.

This finding raises major policy issues. The first one is whether such preferential treatment is justified from a social welfare perspective. The answer depends on the monetization of the larger external costs of ICEVs with respects to BEVs associated with local and global air pollutants, noise, and natural habitats. Such monetization is likely to vary depending on the country's preferences, where the trip take place, the prevailing congestion levels, population density, and so on; a very complex issue outside the scope of this paper.

A second issue is how to reconcile the need to incentivize BEVs with the need to finance the transport infrastructure without overloading the country's state budget. As BEVs substitute ICEVs, the petrol/diesel consumption reduces together with the tax revenues from the fuel consumption. There is a need to find compensation without hampering the uptake of BEVs. Economic theory argues in favor of distance-based taxes, suggesting that the burden of financing a transport infrastructure should be levied on those who use it the most, contributing to congestion in addition to environmental externalities. The actual implementation of such ideas in an effective, efficient, and equitable manner, however, is a complicated issue.

A third consideration is that the differences in the fiscal burden are due to various factors. As shown, some countries opted to set high rates for $CO_2$ and $NO_x$ emissions, some others opted not to use them. Some countries consider the list price or the weight of the car, while others do not, opting for fixed and low fiscal measures. The latter approach is less politically controversial and easier to implement but it promotes the use of a private car, instead of limiting it to induce a modal shift in favor of public transport. Such a consideration suggests to us that each country, having very different starting points—depending on its development level and political preferences—is likely to adopt a different strategy to find a better balance between decarbonizing the passenger sector, balancing out the national accounts, and satisfying the mobility needs.

In light of the above discussion, potential future research areas are, at least, the following. First, there is need to test at a country level whether the preferential treatment towards

BEVs is justified from a social welfare perspective. The results could help the political process and enhance acceptability. Second, it is important to evaluate the timing and the size of the gap between the need to incentivize BEVs and the need to finance the transport infrastructure without overloading the country's state budget. As technology improves and economies of scale in BEV production set in, incentives could be gradually reduced.

**Funding:** This research received no external funding.

**Institutional Review Board Statement:** Not applicable.

**Informed Consent Statement:** Not applicable.

**Data Availability Statement:** Data is available on request from the author.

**Conflicts of Interest:** The author declare no conflict of interest.

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
