# Peer review of "Fiscal Policies on New Passenger Cars in Europe: Implications for the Competitiveness of Electric Cars"

_sustainability, doi:10.3390/su152316407_

Round 1

Reviewer 1 Report

Comments and Suggestions for Authors

I would like to express my gratitude to the author for sharing his work and allowing me to review it. The article presents insightful and relevant considerations, illustrating, among other things, how significantly the approach to the promotion of electric vehicles varies in different European countries, even though they generally agree on the necessity of implementing sustainable transportation solutions.

Please find below my feedback and recommendations for enhancing the article.

1.       The characteristic feature of scientific articles is the inclusion of hypotheses and/or research questions. The presented text lacks these elements.

2.       To facilitate comprehension for readers, scientific articles should include a clearly formulated main purpose for which they were created (main goal of the article). Currently, the introduction and abstract provide information about the content of the article, I would recommend, however, adding a sentence starting with something like, "The main objective of the article is..."

3.       Based on the current title, one might get the impression that the author will be examining the competitiveness of electric cars in comparison to conventional cars, however, most of the discussions revolve around the comparison of electric and hybrid vehicles.

4.       It might be worthwhile to compare the presented results regarding tax burdens in individual countries with the share of electric and hybrid cars in the overall sales of new passenger vehicles.

5.       The sentence "The paper reviewed the fiscal policies applied to new passenger cars in Europe. Line 370" is not precise because most of the discussions presented in the article are related to electric and hybrid cars, not the entire market of passenger vehicles.

6.       In the text, there is a passage that says "the Tesla Model 3 and the Toyota Corolla - chosen as representative of the electric (BEV) and petrol hybrid cars (HEV)." However, the author does not indicate why these two car models were considered representative. As I suspect, this is likely due to their significant market share in electric and hybrid car markets, but the reader may not necessarily be aware of this fact.

7.       The current form of the figures is not very clear due to the very small font size. I'm not sure if the author will be able to improve this given the amount of data presented in the figures, but it's worth trying to increase the readability of the figures.

8.       The phrase "the two car models" appearing in the titles of the figures, I suggest replacing it with, for example, "the electric and hybrid vehicles."

Author Response

First of all, let me express my gratitude for the interesting comments and suggestions provided by the reviewers.

Please find below the reply to each reviewers’ comments.

Reviewer 1

Suggestions for Authors

I would like to express my gratitude to the author for sharing his work and allowing me to review it. The article presents insightful and relevant considerations, illustrating, among other things, how significantly the approach to the promotion of electric vehicles varies in different European countries, even though they generally agree on the necessity of implementing sustainable transportation solutions.

ANSWER. Thanks you for your comments that help me to improve the paper.

Please find below my feedback and recommendations for enhancing the article.

  1. The characteristic feature of scientific articles is the inclusion of hypotheses and/or research questions. The presented text lacks these elements.

ANSWER. Because of the nature of the paper, no hypotheses could be formulated. The research questions explicitly mentioned in the paper are the following:

  1. the level of heterogeneity of the car fiscal policy strategies applied in each country;
  2. their impact on BEV cost competitiveness;
  • how the fiscal tax burden is distributed between BEVs and internal combustion engine cars (ICEVs).
  1. To facilitate comprehension for readers, scientific articles should include a clearly formulated main purpose for which they were created (main goal of the article). Currently, the introduction and abstract provide information about the content of the article, I would recommend, however, adding a sentence starting with something like, "The main objective of the article is..."

ANSWER. Thanks for the advice. I accepted it and formulated it as follows:

“The paper has two main objectives: i) to illustrate and compare the fiscal policies applied in Europe at national level (30 countries are considered); and ii) to estimate their impact of the car acquisition costs and the fiscal burden of two successful cars models - the Tesla Model 3 and the Toyota Corolla - chosen as representative of the electric (BEV) and a petrol hybrid cars (HEV).”

  1. Based on the current title, one might get the impression that the author will be examining the competitiveness of electric cars in comparison to conventional cars, however, most of the discussions revolve around the comparison of electric and hybrid vehicles.

ANSWER. Yes, I thought about this. I opted for formulating the titled as “Fiscal policies on new passenger cars in Europe: implications for the competitiveness of electric cars”, without specifying vs conventional cars or HEVs, for these reasons. 1) The national policies distinguish mainly BEV and all other cars. Rarely, HEVs and Diesel or Petrol cars are considered separately. The CO2 metric is the main instrument to apply a policies. 2) it is true that I selected the Toyota Corolla with the hybrid powertrain. However, what counts for the tax levied are the CO2 emissions, not so much the type of powertrain. More in general, with the definition of the hybrid powertrain (either in the mild of strong version) is somewhat blurred. And the distinction between petrol and hybrid car is not clear cut. That is why I opted for leaving the title open regarding the counterpart to electric cars.

  1. It might be worthwhile to compare the presented results regarding tax burdens in individual countries with the share of electric and hybrid cars in the overall sales of new passenger vehicles.

ANSWER. I thought about that but I refrained from doing it since it opens a new topic which could not be thoroughly discussed this article. It needs a separate econometric analysis.

  1. The sentence "The paper reviewed the fiscal policies applied to new passenger cars in Europe. Line 370" is not precise because most of the discussions presented in the article are related to electric and hybrid cars, not the entire market of passenger vehicles.

ANSWER. I do not fully agree. The first part of the paper reviewed the fiscal policies applied to new passenger cars in Europe, i.e. the criteria on which they are based. The second part applied them to two specific car models, a BEV and an HEV.

  1. In the text, there is a passage that says "the Tesla Model 3 and the Toyota Corolla - chosen as representative of the electric (BEV) and petrol hybrid cars (HEV)." However, the author does not indicate why these two car models were considered representative. As I suspect, this is likely due to their significant market share in electric and hybrid car markets, but the reader may not necessarily be aware of this fact.

ANSWER. Thanks, the following information has been added. ….the Tesla Model 3 and the Toyota Corolla - chosen as representative of the electric (BEV) and a petrol hybrid cars (HEV) since comparable in size and widely popular in their segment. In 2022 in Europe, 129,667 Toyota Corolla (19th ranking most sold car across all segments) and 121,610 Model 3 (second only to the Model Y among BEVs) were registered.

  1. The current form of the figures is not very clear due to the very small font size. I'm not sure if the author will be able to improve this given the amount of data presented in the figures, but it's worth trying to increase the readability of the figures.

ANSWER. OK. It is definitely possible to improve them with the help of the Journal

  1. The phrase "the two car models" appearing in the titles of the figures, I suggest replacing it with, for example, "the electric and hybrid vehicles."

ANSWER. Since the application is model specific, I thought it more correct to refer to the selected models.

Reviewer 2 Report

Comments and Suggestions for Authors

1. In abstract author didnot clearly mentioned about the methodology of this reseach paper. 

2. In Introduction it is not clear what really author wants to find out in this research paper , means research hypotheis or research question is not clear to me. 

3. In introduction author have to clearly tell the motivation to do this research, which is not claer to me. 

4. In this area there are number of literature availabel, but author literature revise section is not enough, need to use more recent literature. 

5. Author could use the following papers - 

Hasan, F., and Islam, M.R. (2022). New energy vehicles from the perspective of market and environment. Journal of Business Strategy, Finance and Management. 4(1), 38-51.

Hasan, F., Bellenstedt, M.F.R.  and Islam, M.R. (2023). The impact of demand and supply disruptions during the Covid-19 crisis on firm productivity. Global Journal of Flexible Systems Management. 24(1), 87-105.

6. Author methodoloy is not clear to me, need a separat section for that.

7. In the anlysis section author need to analysis their findings in mre critically. rather than just simply say what they found. Need to back up their findings using relevant literature.

8. Could provide some table relevant to their findings. 

9. In the conclusion section authir could provide their research limitation and future research. 

Comments on the Quality of English Language

Need minor proofread.

Author Response

First of all, let me express my gratitude for the interesting comments and suggestions provided by the reviewers.

Please find below the reply to each reviewers’ comments.

Reviewer 2

Suggestions for Authors

  1. In abstract author did not clearly mentioned about the methodology of this research paper. 

ANSWER. This paper is not based on a standard methodology that can be described in formal terms. In the second part of the paper, for each country, the relevant metrics have been calculated on the basis of the specific technical and economic characteristics of the two car models. Each countries applies different methodologies, as described in the first part of the paper. However, as suggested by the reviewer, the title of the section has been changed to “The application of fiscal policies to two specific car models: description of the methodology and results”

  1. In Introduction it is not clear what really author wants to find out in this research paper, means research hypothesis or research question is not clear to me.  In introduction author have to clearly tell the motivation to do this research, which is not clear to me. 

ANSWER. Because of the nature of the paper, no hypotheses could be formulated. The research questions explicitly mentioned in the paper are the following:

  1. the level of heterogeneity of the car fiscal policy strategies applied in each country;
  2. their impact on BEV cost competitiveness;
  • how the fiscal tax burden is distributed between BEVs and internal combustion engine cars (ICEVs).
  1. In this area there are number of literature available, but author literature revise section is not enough, need to use more recent literature. Author could use the following papers - 
  • Hasan, F., and Islam, M.R. (2022). New energy vehicles from the perspective of market and environment. Journal of Business Strategy, Finance and Management. 4(1), 38-51.
  • Hasan, F., Bellenstedt, M.F.R.  and Islam, M.R. (2023). The impact of demand and supply disruptions during the Covid-19 crisis on firm productivity. Global Journal of Flexible Systems Management. 24(1), 87-105.

ANSWER. The following papers have been added.

  1. Hasan, S. Assessment of electric vehicle repurchase intention: A survey-based study on the Norwegian EV market. Transp. Res. Interdiscip. Perspect. 2021, 11, 100439.
  2. Lévay, P.Z.; Drossinos, Y.; Thiel, C. The effect of fiscal incentives on market penetration of electric vehicles: A pairwise comparison of total cost of ownership. Energy Policy 2017, 105, 524–533.
  3. Gerlagh, R.; van den Bijgaart, I.; Nijland, H.; Michielsen, T. Fiscal Policy and CO 2 Emissions of New Passenger Cars in the EU. Environ. Resour. Econ. 2018, 69, 103–134.
  4. Lam, A.; Mercure, J.F. Which policy mixes are best for decarbonising passenger cars? Simulating interactions among taxes, subsidies and regulations for the United Kingdom, the United States, Japan, China, and India. Energy Res. Soc. Sci. 2021, 75.
  1. In the analysis section author need to analyze their findings in more critically, rather than just simply say what they found. Need to back up their findings using relevant literature.

ANSWER. Thanks for the suggestion. We added a comparison with Carpenter, G., Antich, A.O. The good tax guide: A comparison of car taxation in Europe; Transport & Environment, Ed.; 2022, which is the study that both in terms of year of analysis and methodology is more similar to this one. We added the following sentences:

  • “Our results are in line with those found by [9]. They underline also that the net tax differential is achieved in some countries by subsidizing BEVs, while in others by imposing a large tax burden on polluting cars. [9] found that this holds true for both compact cars and SUVs.”

  1. Could provide some table relevant to their findings. 

ANSWER. The main results are in the following tables.

  • Figure 4 – Acquisition costs of the two car models
  • Figure 7 – Fiscal burden on the two car models during 8 years

  1. In the conclusion section author could provide their research limitation and future research. 

ANSWER. We changed the title of the last section to “Discussion, limitations of the study and implications for electric vehicle incentive policies”.

Two limitations are highlighted in the paper as follows:

  1. “Such an assumption is evidently a simplification since:
    • it does do not consider that car manufacturers\car retailers might apply different price differentiation strategies across countries to take into account difference in con-sumers preferences and purchasing power;
    • it does not consider that car prices vary over time to respond to the competition or to adjust to cost changes;
    • and, more importantly, the car manufacturers\car retailers might adjust their MSRPs to respond to fiscal policies. For instance, if a certain government grants a subsidy on BEV, they might increase their MSRP to capture some of the fiscal stimulus. Alterna-tively, in some countries regulations are enacted that require car manufacturers\car retailers to supplement the government subsidy with car their own discounts.”
  2. “A caveat is that the paper did not take into account road distance-based tax, the congestion charge, the parking fee or other fees associated with the resale and disposal of the car. Leaving out some of these taxes might be relevant in some cases. For instance, the congestion charge and the parking fee within large cities might further incentivize BEVs adoption (for instance, the congestion charge in Milan and London, or the parking fee in Norway). Distance-based tax taxes, on the contrary, might be set irrespective of the type of powertrain (e.g., the highway toll in Italy).”

Regarding future research, we added the following paragraph:

“In light of the above discussion, potential future research areas are, at least, the fol-lowing. First, there is need to test at country level whether the preferential treatment to-wards BEVs is justified from a social welfare perspective. The results could help the polit-ical process and enhance acceptability. Second, it is important to evaluate the timing and the size of the gap between the need to incentivize BEVs and the need to finance the transport infrastructure without overloading the country’s state budget. As technology improves and economies of scale in BEV production set in, incentives could be gradually reduced.”

Round 2

Reviewer 2 Report

Comments and Suggestions for Authors

Accept in its present form